# Leukocytic Infiltration of Intraductal Carcinoma of the Prostate: An Exploratory Study

**DOI:** 10.3390/cancers15082217

**Published:** 2023-04-09

**Authors:** Mame-Kany Diop, Oscar Eduardo Molina, Mirela Birlea, Hélène LaRue, Hélène Hovington, Bernard Têtu, Louis Lacombe, Alain Bergeron, Yves Fradet, Dominique Trudel

**Affiliations:** 1Centre de Recherche du Centre Hospitalier de l’Université de Montréal (axe Cancer) and Institut du Cancer de Montréal, 900 Saint-Denis, Montréal, QC H2X 0A9, Canada; mame-kany.diop.chum@ssss.gouv.qc.ca (M.-K.D.); mirela.birlea.chum@ssss.gouv.qc.ca (M.B.); 2Department of Pathology and Cellular Biology, Université de Montréal, 2900 Boulevard Édouard-Montpetit, Montréal, QC H3T 1J4, Canada; 3Centre de Recherche du CHU de Québec-Université Laval (axe Oncologie), Hôpital L’Hôtel-Dieu de Québec, 9 McMahon, Québec, QC G1R 3S3, Canada; oscar-eduardo.molina.1@ulaval.ca (O.E.M.); h_larue@videotron.ca (H.L.); helene.hovington@crchudequebec.ulaval.ca (H.H.); bernard.tetu.1@ulaval.ca (B.T.); louis.lacombe@crchudequebec.ulaval.ca (L.L.); alain.bergeron@crchudequebec.ulaval.ca (A.B.); yves.fradet@crchudequebec.ulaval.ca (Y.F.); 4Department of Pathology, CHU de Québec-Université Laval, 11 Côte du Palais, Québec, QC G1R 2J6, Canada; 5Department of Surgery, Université Laval, 2325 rue de l’Université, Québec, QC G1V 0A6, Canada; 6Department of Pathology, Centre Hospitalier de l’Université de Montréal, 1051 Sanguinet, Montréal, QC H2X 0C1, Canada

**Keywords:** prostate cancer, intraductal carcinoma of the prostate, radical prostatectomy, T-lymphocytes, antigen-presenting cells

## Abstract

**Simple Summary:**

Men with a particular type of prostate cancer, called intraductal carcinoma of the prostate (IDC-P), are more likely to die from their cancer than men without IDC-P. No researchers have yet compared the immune infiltrate of IDC-P to the immune infiltrate of prostate cancer. In this study, we quantified immune cells specifically in IDC-P and compared the cell densities in IDC-P to those in the adjacent cancer, tumor margins and benign tissues. We found that the immune infiltrate of IDC-P was generally reduced compared to the surrounding tissues, especially regarding antigen-presenting cells. Following validation in larger cohorts, the characterization of the immune microenvironment of IDC-P could allow a better understanding of the immune response to lethal prostate cancer and enable the development of new therapies.

**Abstract:**

Intraductal carcinoma of the prostate (IDC-P) is an aggressive histological subtype of prostate cancer (PCa) detected in approximately 20% of radical prostatectomy (RP) specimens. As IDC-P has been associated with PCa-related death and poor responses to standard treatment, the purpose of this study was to explore the immune infiltrate of IDC-P. Hematoxylin- and eosin-stained slides from 96 patients with locally advanced PCa who underwent RP were reviewed to identify IDC-P. Immunohistochemical staining of CD3, CD8, CD45RO, FoxP3, CD68, CD163, CD209 and CD83 was performed. For each slide, the number of positive cells per mm^2^ in the benign tissues, tumor margins, cancer and IDC-P was calculated. Consequently, IDC-P was found in a total of 33 patients (34%). Overall, the immune infiltrate was similar in the IDC-P-positive and the IDC-P-negative patients. However, FoxP3^+^ regulatory T cells (*p* < 0.001), CD68^+^ and CD163^+^ macrophages (*p* < 0.001 for both) and CD209^+^ and CD83^+^ dendritic cells (*p* = 0.002 and *p* = 0.013, respectively) were less abundant in the IDC-P tissues compared to the adjacent PCa. Moreover, the patients were classified as having immunologically “cold” or “hot” IDC-P, according to the immune-cell densities averaged in the total IDC-P or in the immune hotspots. The CD68/CD163/CD209-immune hotspots predicted metastatic dissemination (*p* = 0.014) and PCa-related death (*p* = 0.009) in a Kaplan–Meier survival analysis. Further studies on larger cohorts are necessary to evaluate the clinical utility of assessing the immune infiltrate of IDC-P with regards to patient prognosis and the use of immunotherapy for lethal PCa.

## 1. Introduction

Despite the tremendous advances made in cancer immunotherapy in the last decade [1,2,3], the efficacy of immunological approaches remains modest for advanced prostate cancer (PCa) [4,5,6]. Sipuleucel-T, an autologous cellular-immunotherapy vaccine based on antigen-presenting cells (APC), was approved by the Food and Drug Administration (FDA) in 2010 for the treatment of advanced PCa; Sipuleucel-T offers a slight 4-month overall survival benefit compared to placebo in men with metastatic castration-resistant PCa (mCRPC) [7]. More recently, a checkpoint inhibitor, pembrolizumab, was approved by the FDA for cancers with high microsatellite instability or mismatch-repair deficiency, regardless of the tumor origin [8]. However, barely 3% of men with advanced PCa have such alterations [9]. Other immunotherapy drugs are yet to be approved.

Although ongoing clinical trials are exploring immunotherapy-based combinatorial strategies for advanced PCa, it is crucial to better understand the immune response associated with PCa to better identify men who could benefit from these approaches. Most studies evaluating the immune infiltrate of PCa focused on T lymphocytes and found that a higher expression of tumor-infiltrating lymphocytes (TILs) is associated with worse PCa prognosis [10,11,12,13,14,15].

Intraductal carcinoma of the prostate (IDC-P) is an aggressive histologic entity of PCa associated with poor prognosis [16]. It is characterized by the proliferation of malignant prostatic epithelial cells in pre-existing prostatic ducts [17]. Current evidence suggests that most IDC-P arises from the retrograde invasion of conventional prostatic adenocarcinoma into prostatic ducts [18,19,20]. In a systemic review of 38 PCa cohorts, Porter et al. found that the prevalence of IDC-P increased from 2.1% in low-risk PCa patients to 36.7% in high-risk PCa patients, even reaching 56.0% in patients with metastatic disease [21]. Accordingly, men with IDC-P tend to have more advanced PCa, as shown by their higher pathologic (p) T stages and higher Gleason scores [22,23,24]. Furthermore, IDC-P still independently predicts biochemical recurrence (BCR) and the development of metastasis and CRPC [18,21,25,26]; furthermore, it has been linked to poor responses to treatment and worse PCa-specific and overall survival [27,28,29,30,31,32,33,34]. There is currently no consensus on the clinical management of IDC-P [35,36,37].

Because of its association with lethal PCa and its resistance to current therapies, we believe that the immune infiltrate of IDC-P is distinct from that in surrounding tissues, and that its evaluation will allow a better understanding of the microenvironment of advanced PCa. Here, we show that the immune infiltrate of IDC-P is different from that found in the adjacent invasive carcinoma. In addition, we found that the immune infiltrate in IDC-P can be categorized as “cold”, “intermediate” or “hot”, depending on the immune-cell densities averaged in the total IDC-P or in IDC-P-immune hotspots. We found that the patients with “hot” CD68-, CD163- and CD209-immune hotspots experienced shorter metastasis-free and PCa-specific survival.

## 2. Materials and Methods

### 2.1. Patients and Ethics

Radical prostatectomy (RP) specimens from 96 men with locally advanced hormone-naïve PCa who underwent surgery between 1996 and 1998 at the CHU de Québec-Université Laval were included in this study. Locally advanced PCa was defined as either pT3- or pT4-stage PCa or pT2-stage PCa with positive margins. Each participant signed an informed consent form to participate at the local cancer biobank (URO-1 biobank), allowing the use of their tissues and clinical and pathological data for cancer research. This study was conducted following approval by the CHU de Québec-Université Laval Research Ethics Committee (research project 2012-1059) and the CHUM Research Ethics Committee (research project MP-02-2018-7450).

### 2.2. Selection of the Representative Blocks and Identification of IDC-P

Hematoxylin and eosin (H&E) slides were reviewed to select one representative formalin-fixed paraffin-embedded (FFPE) tissue block per patient. The corresponding H&E slide had to be representative of the grade group (GG) assigned at diagnosis, containing between 30% and 70% of cancer and as small a number of calcifications as possible; furthermore, the selected block had to allow at least eight additional tissue sections.

The presence of IDC-P was assessed on the selected H&E slides by one trained observer (M.-K.D.) and a pathologist with expertise in IDC-P (D.T.). Intraductal carcinoma of the prostate was identified when cancer cells invaded the lumen of obvious pre-existing ducts to form dense-to-solid cribriform patterns or loose patterns, provided that either marked pleomorphism, frequent mitotic activity, comedonecrosis or abnormally large nuclei were present, as defined by Guo and Epstein [18].

### 2.3. Immunohistochemistry

Consecutive 5-µm-thick sections of the selected FFPE block were cut, dried overnight at 37 °C and deparaffinized. Heat-induced antigen retrieval was performed using a PT Link (Dako, Burlington, ON, Canada) at 92 °C for 20 min in citrate buffer, pH 6.1 (EnVision^TM^ FLEX, K8004, Dako), for CD3, CD45RO, CD68 and CD209 stainings or Tris/EDTA, pH 9 (EnVision^TM^ FLEX, K8005, Dako), for CD8, FoxP3, CD163 and CD83 stainings. Slides were then incubated for 10 min in 3% hydrogen peroxide. Using the IDetect Super Stain Horseradish Peroxidase (HRP) polymer kit (ID Labs, London, ON, Canada), slides were first incubated for 10 min with Super Block blocking buffer and then with primary antibodies at room temperature (Table 1). With washes between each step, slides were then incubated for 30 min with HRP-polymer-conjugated antibodies and for 5 min with 3,3′-diaminobenzidine (DAB) tetrahydrochloride solution. Lastly, slides were counterstained with hematoxylin, dehydrated and mounted with MM 24 low-viscosity mounting medium (Leica Microsystems, Hurham, NC, USA).

### 2.4. Quantification of Immunohistochemistry Staining

The immunohistochemistry (IHC) slides were scanned using a Nanozoomer whole-slide scanner (Hamamatsu, Bridgewater, NJ, USA).

For each slide, 10 high-power fields at 200× magnification (0.460 mm^2^), or less when less tissue was available, were randomly selected in isolated benign areas (containing ducts), tumor margins (areas on the periphery of the tumor) and within the cancer. The number of positive cells in each area was then calculated using one of the two following methods: manual use of the NDP.view2 software (Hamamatsu Photonics) by two trained observers (O.E.M. and H.L.); or manually by one trained observer (O.E.M) and in a semi-automatic fashion through the Calopix software (TRIBVN Healthcare, Châtillon, France). Ten percent of the slides were reviewed by an experienced pathologist (B.T.).

Positive cells in IDC-P were quantified using the NDP.view2 software (Hamamatsu Photonics). On slides containing IDC-P, each individual IDC-P lesion was encircled using the freehand-region tool in NDP.view2, and positive cells were manually counted by a trained observer (M.-K.D.) to obtain the number of positive cells/mm^2^ for each IDC-P lesion and the mean number of positive cells/mm^2^ in the total IDC-P of the slide. Immune hotspots were defined as the highest number of positive cells/mm^2^ in IDC-P lesions with an area greater than the median area of all IDC-P lesions on the slide. Slides were reviewed by an experienced pathologist (D.T.).

The observers were blinded to the clinical outcomes of the patients.

### 2.5. Clinical-Data Collection and Endpoints

Age, pre-operative prostate-specific antigen (PSA) serum concentrations, pT stage, modified Gleason grading system/GG grading [17,38] and margin status were collected from patient files. Clinical follow-up data were collected from the day of RP. Biochemical recurrence was defined as two consecutive PSA test results over 0.3 ng/mL after RP, one PSA test result over 0.3 ng/mL after post-operative treatment or any increase in serum PSA that required post-operative treatment by a urologist. Castration-resistant prostate cancer was defined as disease progression, including rise in serum PSA and development of metastases, despite castrate levels of serum testosterone [39]. Definitive androgen deprivation therapy (ADT) was defined as any hormonotherapy that was given to a patient as primary treatment (not neoadjuvant or adjuvant) after suspected or confirmed disease progression.

### 2.6. Statistical Analyses

Data analyses were conducted using IBM SPSS Statistics for Windows, version 28.0 (Armonk, NY, USA: IBM Corp.), according to the REMARK guidelines [40]. Descriptive statistical analysis for quantitative variables used mean, median, standard deviation, standard error, inter-quartile range (IQR) and, for qualitative variables, percentage. Univariate analyses were performed using the independent-samples *t*-test, Mann–Whitney *U* test, Welch’s *t*-test, Pearson’s chi-square test, Fisher’s exact test and the paired-samples sign test. To evaluate the correlation between cell densities, we performed Pearson correlations [41]. The Benjamini–Hochberg procedure was used to control for multiple comparisons with a 10% false-discovery rate. Time to BCR, metastasis, CRPC, definitive ADT, PCa-specific death and overall survival were measured from the date of RP until the date of the event or last follow-up date; these events were evaluated using the Kaplan–Meier method and log-rank test. Statistical significance was established for two-sided *p*-values < 0.05.

Bar charts were generated in Excel (Microsoft Corporation, version 18.2301.1131.0). Survival curves were created in IBM SPSS Statistics. Other charts were built using R (R Core Team, 2022, Vienna, Austria) and RStudio (RStudio Team, 2022, Boston, MA, USA). The ggplot2 [42] and ggbreak [43] packages were used to generate jitter plots and parallel coordinates charts, while the plot function was used to generate correlation matrix and the heatmap.2 function of the gplots package [44] was used for heatmap and hierarchical clustering.

## 3. Results

### 3.1. Clinicopathological Characteristics of Patients

The clinicopathological characteristics of the 96 patients included in this study are presented in Table 2. A total of 33 patients (34%) had IDC-P on their selected block. The patients in the IDC-P-negative group and in the IDC-P-positive group had similar median follow-ups of 15.8 years and 14.3 years (*p* = 0.105), respectively. As expected, the presence of IDC-P was associated with higher pT stages (*p* < 0.001) and GGs (*p* = 0.001). Indeed, the pT3b and pT4 stages were found in 58% (19/33) of the patients in the IDC-P-positive group compared to 21% (13/63) of the patients in the IDC-P-negative group. Furthermore, high grades (GGs 4 and 5) were diagnosed in 36% (12/33) of the patients from the IDC-P-positive group compared to 10% (6/63) of the patients from the IDC-P-negative group. Accordingly, the men with IDC-P had higher rates of lymph-node involvement (42% vs. 21%, *p* = 0.032). Moreover, IDC-P was associated with the development of BCR (73% vs. 35%, *p* < 0.001), CRPC (36% vs. 8%, *p* < 0.001) and metastases (39% vs. 6%, *p* < 0.001). In addition, IDC-P was associated with higher rates of PCa-specific death (33% vs. 5%, *p* < 0.001) and overall death (73% vs. 41%, *p* = 0.005). No statistically significant differences were found in the rates of positive margins (*p* = 1.000) and lymphovascular invasion (*p* = 0.332) between the two groups.

### 3.2. Quantification of CD3⁺, CD8⁺, CD45RO⁺, FoxP3⁺, CD68⁺, CD163⁺, CD209⁺ and CD83⁺ Cells in Benign Tissues, Margins, Cancer and IDC-P

The immune cells were detected through the immunohistochemical staining of CD3 (T lymphocytes), CD8 (cytotoxic T lymphocytes), CD45RO (memory T cells), FoxP3 (regulatory T cells), CD68 (macrophages), CD163 (M2-type macrophages), CD209 (immature dendritic cells) and CD83 (mature dendritic cells) (Figure 1). The FoxP3 expression was localized in the nuclei of the T cells, whereas CD3, CD8, CD45RO, CD209, CD83, CD68 and CD163 expression was localized in the cell membrane. These eight markers were quantified in isolated benign tissues, cancer margins and within the whole cancer area (regions with invasive carcinoma selected randomly and independently of the presence of IDC-P).

The average cell densities (mean number of positive cells per mm²) ranged from 3 to 836 for CD3, 4 to 163 for CD8, 1 to 238 for CD45RO, 0 to 45 for FoxP3, 1 to 335 for CD68, 0 to 280 for CD163, 0 to 59 for CD209 and 0 to 40 for CD83 (Figure 2). The cell densities tended to be higher in the margins compared to the benign and cancer compartments, except for the CD68⁺ macrophages, which tended to be more abundant in the cancer compartment and the CD83⁺ dendritic cells in benign tissues (*p* > 0.05). Furthermore, no significant differences were seen in the mean ranks of the immune cells between the IDC-P-negative and the IDC-P-positive samples in the same compartment, excluding the FoxP3⁺ cells, which were more abundant in the margins of the IDC-P-positive samples (mean ranks: 55.0 vs. 41.0, *U* = 608, *p* = 0.016) (Figure 2).

Next, to further evaluate the inflammation associated with IDC-P, we specifically assessed the immune infiltrate within the IDC-P areas (Figure 1). The mean area of IDC-P per slide was 6.5 mm^2^ and the median was 3.2 mm^2^, with a minimum area of 0.1 mm^2^ and a maximum area of 25.6 mm^2^. The immune infiltrate of IDC-P varied broadly between samples for CD3, CD8, CD45RO and CD68, with median (range) counts of 48 (6–244)/mm^2^, 39 (5–174)/mm^2^, 38 (8–150)/mm^2^ and 44 (12–211)/mm^2^, respectively. Only a small portion of macrophages in the IDC-P seemed to express CD163, with a median count of 4 (0–51)/mm^2^. By contrast, FoxP3^+^, CD209^+^ and CD83^+^ cells were scarce in the IDC-P, ranging from 0 to 15 cells/mm^2^, 0 to 4 cells/mm^2^ and 0 to 4 cells/mm^2^, respectively.

When we compared the cell densities in the IDC-P to the other compartments, we found significant differences. Paired-samples sign tests were conducted to assess the variations in the mean number of immune cells within the same slide and the results are presented in Figure 3. Regardless of the compartment, benign, margins or cancer, CD163⁺ macrophages (*p* < 0.001 for all three compartments) and CD209⁺ (*p* < 0.001 for benign and margins and *p* = 0.002 for cancer) or CD83⁺ (*p* < 0.001 for benign and margins and *p* = 0.013 for cancer) dendritic cells were typically less abundant in the IDC-P (Figure 3). Furthermore, CD68⁺ macrophages (*p* = 0.036 for margins and *p* < 0.001 for cancer) and FoxP3⁺ regulatory T lymphocytes (*p* < 0.001 for both) were typically less abundant in the IDC-P compared to margins and cancer (Figure 3b,c). Notably, except for the CD68^+^ macrophages, which were generally more abundant in the cancer than in the tumor margins, leucocytes were typically more abundant in the tumor margins than in the cancer (Figure 3b,c and Appendix A).

Interestingly, no significant differences were found between the variations in the mean number of T lymphocytes (CD3⁺, CD8⁺, CD45RO⁺ and FoxP3⁺) and CD68⁺ macrophages in the IDC-P compared to the benign tissues (Figure 3a). Indeed, as shown in Figure 3, the decrease in the densities of the T cells in the IDC-P was more balanced, with a considerable number of patients having more positive cells in IDC-P tissues compared to benign tissues. Furthermore, most of the patients had more CD3⁺ (*p* = 0.006) and CD8⁺ (*p* = 0.004) T cells and tended to have more CD45RO⁺ T cells (*p* = 0.052) in the margins compared to IDC-P (Figure 3b), while the CD3⁺, CD8⁺ and CD45RO⁺ cell densities were less often decreased in IDC-P tissues compared to cancer or benign tissues (Figure 3a,c), resulting in a non-significant difference between the two compartments. Notably, we observed CD45RO^+^ cell clusters surrounding the IDC-P tissues in 56% (15/27) of the cases and the number of CD45RO^+^ cells/mm^2^ in the IDC-P increased with the presence of surrounding CD45RO^+^ cell clusters (mean CD45RO^+^/mm^2^ ± SD: 63 ± 45 vs. 32 ± 19, *p* = 0.025).

Parallel coordinate plots showing the changes in cell density for each IDC-P-positive patient in the general cancer or in the margins compared to the other compartments are available in the Appendix A.

### 3.3. Correlation between Immune-Cell Densities in IDC-P

Unsurprisingly, we found a strong correlation between the CD3⁺-T-lymphocyte densities and those of the CD8⁺ cytotoxic T lymphocytes (r = 0.90, *p* < 0.001), CD45RO⁺ memory T lymphocytes (r = 0.87, *p* < 0.001) and FoxP3⁺ regulatory T lymphocytes (r = 0.72, *p* < 0.001) in the IDC-P tissues (Figure 4a). The CD8⁺ and CD45RO⁺ T-cell densities were also strongly correlated in the IDC-P tissues (r = 0.80, p < 0.001). The CD163⁺-M2-type-macrophage densities were not significantly correlated with the CD68⁺ cells (r = 0.27, *p* = 0.196; Figure 4b), nor were the CD209⁺ and CD83⁺ dendritic cells (r = 0.13, *p* = 0.514; Figure 4c). Interestingly, the T-lymphocyte densities in the IDC-P tissues were moderately correlated with the CD163⁺-M2-macrophage densities (r = 0.59, *p* = 0.002; Figure 4b) and, albeit to a lesser extent, with those of the CD83⁺ mature dendritic cells (r = 0.48, *p* = 0.013) (Figure 4c) and CD68⁺ macrophages (r = 0.41, *p* = 0.047; Figure 4b), but not with those of the CD209^+^ immature dendritic cells (r = 0.02, *p* = 0.938; Figure 4c).

### 3.4. Clusterization of IDC-P-Positive Patients According to Immune-Cell Densities

The mean numbers of positive cells per mm^2^ according to the patients’ outcomes after five and 10 years of follow-up are presented in Figure 5. No significant differences were seen in the mean ranks of the leukocytes in the IDC-P tissues between patients with different outcomes, except for CD68^+^ cells, which were more abundant in the men who died of their PCa (mean ranks: 18.0 vs. 10.2, *U* = 21, adjusted *p* = 0.047) and the men who died regardless of cause (mean ranks: 17.8 vs. 12.7, *U* = 40, adjusted *p* = 0.047) (Figure 5b).

We further explored the immune infiltrate of IDC-P by examining the immune signatures of each patient according to the average immune-cell densities in the total IDC-P. Three patients were excluded due to the absence of more than three IHC slides. The mean numbers of positive immune cells per mm^2^ in the IDC-P tissues were standardized using z-score normalization to generate a heatmap representing the unsupervised hierarchical clustering of both the rows and the columns. Two main groups were identified according to the expression of the eight immune-cell markers: men with a low expression of immune markers, hereafter referred to as “cold” IDC-P; and men with a higher expression of immune markers, hereafter referred to as “hot/intermediate” IDC-P (Figure 6a).

The median follow-up of the 30 patients with IDC-P was 164 months (IQR: 89–209). We generated Kaplan–Meier curves to explore the association between the expression of the immune markers and the clinical outcomes, according to the two groups: “cold” or “hot/intermediate” IDC-P (Figure 6b). Although, in this small cohort, statistical significance was not reached, there was a trend toward better prognoses for patients with immunologically “cold” IDC-P. Only 19% of the men with “hot/intermediate” IDC-P were BCR-free after 10 years of follow-up compared to twice as many men with “cold” IDC-P. The need for definitive ADT, which is an indication of recurrence, was also lower in the “cold” IDC-P group (10-year definitive ADT-free survival rates: 70% for the “cold” group vs. 43% for the “hot/intermediate” group). At ten years of follow-up, 32% of the patients with “cold” IDC-P developed CRPC, 25% developed metastasis and 25% died of their PCa compared to 35%, 41% and 31% of the patients with “hot/intermediate” IDC-P, respectively. Finally, under one third of the men with “cold” IDC-P died within 10 years of follow-up, compared to nearly half of the patients with “hot/intermediate” IDC-P.

We then examined the association between the cell densities in the IDC-P-immune hotspots and prognosis. The dependence of the mean numbers of cells per mm^2^ on the outcomes after five and 10 years of follow-up are illustrated in Figure 7. No significant differences were seen in the T-cells densities between the groups (Figure 7a). However, higher densities were seen in the hotspots in the patients who died of their cancer or of other causes within five years for CD68^+^ macrophages (mean ranks: 24.3 vs. 12.7, *U* = 5, adjusted *p* = 0.035 and mean ranks: 23.8 vs. 13.0, *U* = 11, adjusted *p* = 0.035, respectively) or within 10 years for CD163^+^ macrophages (mean ranks: 18.7 vs. 9.9, *U* = 16, adjusted *p* = 0.012 and mean ranks: 20.6 vs. 10.6, *U* = 27, adjusted *p* = 0.006, respectively) (Figure 7b). Greater CD163^+^-cell densities were also associated with the development of metastasis during the first 10 years of follow-up (mean ranks: 17.3 vs.10.1, *U* = 26, adjusted *p* = 0.035) (Figure 7b). Furthermore, higher CD209^+^-cell densities predicted the need for definitive ADT within five (mean ranks: 21.9 vs. 12.4, *U* = 29, adjusted *p* = 0.005) and 10 years (mean ranks: 18.4 vs. 11.0, *U* = 40, adjusted *p* = 0.031) (Figure 7b).

The immune signatures of the patients were generated using the cell densities of the three markers significantly associated with prognosis in the IDC-P-immune hotspots CD68, CD163 and CD209. One additional patient was excluded due to the absence of more than one IHC slide. Again, the cell densities in the IDC-P tissues were standardized using z-score normalization to generate a heatmap representing unsupervised hierarchical clustering. Two main groups were identified according to the expression of the three immune-cell markers in the hotspots: men with a low or intermediate expression of immune markers, constituting the “cold/intermediate” group; and men with a higher expression of immune markers, constituting the “hot” group, largely due to the presence of macrophages (Figure 8a).

We also generated Kaplan–Meier curves to explore the association between the expression of CD68, CD163 and CD209 and the clinical outcomes, according to the two groups: “cold/intermediate” or “hot” IDC-P hotspots (Figure 8b). The median follow-up of the 29 patients was 172 months (IQR: 96–209). Higher expressions of immune APC still tended to be associated with worse prognoses. Despite our small number of patients, we found that the median time to metastasis (number of events = 11) and PCa-related death (number of events = 10) was significantly shorter in the patients with “hot” APC hotspots (65 months, 95% confidence interval (CI): 0–184 vs. median survival not yet reached in patients in the “cold/intermediate” group, with *p* = 0.014 for metastasis; and 98 months, 95% CI: 0–262 vs. median survival not yet reached in patients in the “cold/intermediate” group, with *p* = 0.009 for PCa-specific death) (Figure 8b).

When clinico-pathological characteristics of patients with immunologically “cold”, “intermediate” or “hot” IDC-P were compared, extraprostatic extension was the only characteristic that was found to be more frequent in the “cold” total-IDC-P group than in the “hot/intermediate” total-IDC-P group (88% vs. 54%, *p* = 0.049; Table 3).

Notably, four men were in both the “hot/intermediate” IDC-P group, according to the cell densities in the total IDC-P (*n* = 13) and in the “hot” IDC-P group, according to the hotspots (*n* = 7). In addition, three out of the seven men (43%) in the latter group had comedonecrosis in at least one of their slides (CD68: one slide, comedonecrosis in 3/196 affected ducts; CD163: one slide, comedonecrosis in 2/78 affected ducts, CD209: two slides, comedonecrosis in 1/190 and 1/111 affected ducts) and only CD68^+^ cells were found in the necrotic region of one slide.

## 4. Discussion

Intraductal carcinoma of the prostate is linked to metastasis and lethality in PCa [16]. In this exploratory study, we are the first to explore the immune infiltrate of IDC-P, an aggressive histologic subtype of PCa characterized by the growth of cancer cells within the prostatic ducts. We found that the immune microenvironment of IDC-P is distinct from that of the adjacent invasive carcinoma. Specifically, CD163⁺ M2-type macrophages and CD68⁺ macrophages, FoxP3⁺ T cells and CD209⁺ and CD83⁺ dendritic cells are often less abundant in IDC-P than in the adjacent invasive PCa. Furthermore, the IDC-P patients were separated into groups, according to the immune-cell densities in the total IDC-P or in the IDC-P-immune hotspots. Regardless of category, patients with “colder” IDC-P tend to have better survival rates than patients with more leucocytes in IDC-P. Moreover, we found an association between the development of metastasis and PCa-specific death in men with higher expressions of CD68, CD163 and CD209 in IDC-P-immune hotspots.

Until now, TILs in PCa were assessed in areas containing conventional invasive carcinoma, or without addressing the presence of IDC-P. However, IDC-P alone has been associated with shorter disease-specific survival in men with localized PCa and men with high-risk PCa [16], making it an attractive lesion in which to study the immune response associated with lethal PCa. We found that the immune infiltrates of IDC-P were mainly reduced compared to randomly selected regions in the benign tissues, the tumor margins and, most importantly, in the adjacent invasive cancers. An explanation for this could be related to the association of IDC-P with hypoxia [45]. Indeed, hypoxic environments are believed to be highly immunosuppressive [46]. Furthermore, in our study, the immunosuppressive environment in IDC-P was even more pronounced in association with APC (dendritic cells and macrophages). These findings suggest that the immune environment of IDC-P is different from those in the adjacent carcinoma and other surrounding tissues, and might be better suitable for the study of the tumor microenvironment in the context of aggressive PCa.

Most studies assessing tumor-infiltrating immune cells in PCa and clinical outcomes suggested that the high expression of TILs is associated with poor prognosis. The first studies on this topic date back to the 1990s, when Vesalainen et al. [47] and then Irani et al. [48] visually semi-quantified TILs in H&E-stained PCa tissues. The first group (*n* = 325) found that a low expression of TILs was associated with the development of distant metastasis (univariate analysis: *p* = 0.016) and with lethal disease in patients without metastasis at diagnosis (multivariate hazard ratio (HR): 0.67, CI: 0.47–0.95, *p* = 0.012) and patients with organ-confined disease and without metastasis at diagnosis (multivariate HR: 0.44, 95% CI: 0.23–0.84, *p* = 0.041) [47]. The second group (*n* = 161) found that a high expression of stromal TILs was associated with an increased risk of post-RP BCR (multivariate relative risk (RR): 2.35, 95% CI: 1.08–5.08, *p* = 0.03) [48].

From the 2000s onwards, the quantification of leucocytes on IHC-stained PCa tissues was performed, with most studies focusing on T lymphocytes. Ness et al. [10] found that a high expression of CD8⁺ in the epithelial compartment and in the total tumor was an independent predictor of shorter BCR-free survival in tissue microarrays (TMAs) from 535 RP specimens (multivariate HR: 1.45, CI 95%: 1.03–2.03, *p* = 0.032 and multivariate HR: 1.57, CI 95%: 1.13–2.17, *p* = 0.007, respectively). However, in their cohort of 11 RP and 68 transurethral-resection-tissue specimens, McArdle et al. [11] noted that an increased density of CD4⁺ cells, but not of CD8⁺ cells, was independently associated with poorer disease-specific survival (multivariate HR: 2.29, 95% CI: 1.25–4.22, *p* = 0.008). Using a digital approach, Richardsen et al. [12] observed that men with metastatic PCa (*n* = 32) had a higher expression of CD3⁺ T cells in the epithelial (*p* = 0.007) and stromal (*p* < 0.0001) compartments than men with non-metastatic disease (*n* = 27). Kärjä et al. [13] added B lymphocytes in their study and found that high expressions of CD4⁺, CD8⁺ and CD20⁺ were independent predictors of shorter BCR-free survival in TMAs from 188 RP specimens (multivariate HR: 0.18, 95% CI: 0.07–0.44, *p* = 0.012). Furthermore, Flammiger et al. [49] found that very low or very high expressions of CD3⁺ cells were associated with shorter BCR-free survival compared to the intermediate expression of CD3⁺ cells in a large cohort of TMAs collected from 2144 RP specimens (univariate analysis: *p* = 0.0188), while CD20⁺-cell density was not associated with prognosis. The same researchers [50] then suggested that a high expression of FoxP3⁺ T cells was associated with reduced BCR-free survival in a univariate analysis (*p* = 0.0151), but this result was not upheld by their multivariate analysis (*n* = 1463). Similarly, Davidsson et al. [15] observed in their TMA cohort that a high expression of regulatory T lymphocytes increased the odds of dying of PCa by 12% (multivariate odds ratio: 1.12; 95% CI: 1.02–1.23), but that this was not the case with helper or cytotoxic T cells (*n* = 663). More recently, Kaur et al. [14] used an automated quantification method to assess CD3⁺, CD8⁺ and FoxP3⁺ lymphocytes in a TMA cohort comprising 312 PCa patients, including 212 (68%) African-American patients. With BCR and the development of metastasis as endpoints, only high densities of FoxP3⁺ cells (top tertile) remained significantly associated with an increased risk of metastasis according to the researchers’ multivariate analysis (HR: 12.89, 95% CI: 1.59–104.40, *p* = 0.02).

Very few studies examined APC in PCa tissues. In the aforementioned study by Richardsen et al. [12], higher percentages of CD68⁺ cells were seen in the epithelial and stromal primary-tumor compartments of non-metastatic cancers compared to metastatic cancers (48% vs. 28%, *p* = 0.029 in the epithelial compartment; 54% vs. 14%, *p* = 0.008 in the stromal compartment, respectively) (*n* = 59). Moreover, Comito et al. [51] focused on CD68^+^ and CD163^+^ macrophages in hotspots identified in RP specimens from 93 patients with clinically localized PCa. They found that M1 macrophages were more abundant in organ-confined diseases upon final evaluation compared to cancers that presented with extraprostatic extension (≈18 cells/mm^2^ vs. 5 cells/mm^2^). In contrast, M2 macrophages were associated with extraprostatic extension (multivariate RR: 0.30, 95% CI: 0.09–0.89, *p* = 0.03). Patients with a prevalence of M2 macrophages also tended to have shorter BCR-free survival than patients with a prevalence of M1 macrophages, but statistical significance was not reached. In addition, Calagua et al. [52] found a correlation between CD8^+^ T cells and APC niches (R^2^ = 0.57, *p* = 0.0001) through multiplex immunofluorescence, suggesting an important role of APC in T-cell response (*n* = 20).

In our study, we found a correlation between T cells and APC in IDC-P, particularly with M2 macrophages (r = 0.59, *p* = 0.002) and mature dendritic cells (r = 0.48, *p* = 0.013). In contrast, no correlation was found between T cells and immature dendritic cells. Despite the immunosuppressive microenvironment of IDC-P, we separated our IDC-P patients into groups according to their immune-cell densities: patients with immunologically “cold” IDC-P and patients with immunologically “hot” IDC-P. Probably due to the small number of patients with IDC-P and, hence, the limited statistical power of our study, we did not find any significant differences in survival between the men with globally “cold” IDC-P and the men with globally “hot/intermediate” IDC-P. However, when we examined the immune hotspots in IDC-P, we found that the CD68^+^ and CD163^+^ macrophages and CD209^+^ immature dendritic cells were associated with poor prognosis. Immune hotspots have been studied in breast cancer, among others, in which they were found to be associated with better prognosis in estrogen-receptor-negative tumors [53], but were also associated with poorer prognosis in estrogen-receptor-positive tumors [54]. In our cohort, CD68/CD163/CD209-immune hotspots predicted progression to metastatic disease and cancer-specific survival. Altogether, and in accordance with most PCa studies, our results tend to show that increased IDC-P infiltration is associated with poorer prognosis.

The reasons why higher immune-cell densities are associated with worse prognoses in PCa are still not understood. They could be related to the tumors themselves, as well as the tumor microenvironment. Some studies sought explanations by examining alterations in gene expression. Amongst these studies, Kaur et al. [14] showed that higher T-lymphocyte densities were associated with ERG expression (median: 309 vs. 188 CD3^+^ T cells/mm^2^; *p* = 0.0004) and PTEN loss (median: 317 vs. 192 CD3^+^ T cells/mm^2^; *p* = 0.001). Similarly, Calagua et al. [52] found that the deletion of *BRCA* and/or *RB1* was more frequent in their subset of 11 immunogenic patients than in the localized PCa data obtained from the TCGA Firehose legacy (*BRCA2* deletion: *p* = 0.053, *RB1* deletion: *p* = 0.017, co-deletion of *BRCA2*/*RB1*: *p* = 0.053, focal co-deletion of *BRCA2*/*RB1*: *p* = 0.039; *n* = 489). Both ERG expression and PTEN loss [55,56], as well as the co-loss of BRCA2/RB1, have been associated with aggressive forms of PCa [57]. Interestingly, a few limited studies linked BRCA^mut^ to IDC-P [16].

Our study has limitations that warrant discussion. The main limitation is the small number of patients with IDC-P in our cohort (*n* = 33). However, IDC-P is often focal, with more than 75% of specimens with IDC-P harboring IDC-P in less than 5% of the full tumor volume [58]. Furthermore, the evaluation of the inflammatory infiltrate of adjacent invasive carcinoma, tumor margins and benign glands requires spatial localization, which cannot be provided by biopsies and TMAs. These limitations in sample availability could explain why the immune infiltrate of IDC-P have not yet been described. Our findings must still be replicated in larger independent cohorts, but our work was essential to begin to unravel the signification of the immune infiltrate in IDC-P. Larger cohorts will also allow to separately evaluate men with immunologically “intermediate” IDC-P and the performance of multivariate analyses. Moreover, we only examined one representative block per patient, meaning that the patients in the IDC-P-negative group could have had IDC-P in other tissue blocks. However, despite this, IDC-P was still associated with poor prognoses and adverse pathological features in our cohort [16]. Furthermore, we performed single-color IHC to characterize the immune cells in PCa instead of multiplex immunofluorescence, which would have permitted us to assess the colocalization of the markers and quantify all the markers on the same slides, in addition to allow for the confirmation of the presence of basal cells around IDC-P. In addition, we examined T lymphocytes and macrophages, but other key immune cells [59], such as B lymphocytes, natural killer cells, classical neutrophiles and monocytes and myeloid-derived suppressor cells [60] should be included in further research. Futures studies will incorporate these leucocytes in addition to other key actors in the immune response. However, our study highlights the importance of evaluating IDC-P in the study of the immune environment of PCa.

## 5. Conclusions

In conclusion, we found that the immune infiltrate of IDC-P is different from that in the adjacent invasive carcinoma, while the overall immune infiltration is not affected by the IDC-P status. Antigen-presenting cells are particularly less abundant in IDC-P compared to cancer in general. Moreover, IDC-P can be classified as immunologically “cold” or “hot”, depending on the immune-cell densities. In this study, these groups were associated with different clinical outcomes, and CD68/CD163/CD209-immune hotspots predicted progression to metastatic disease and cancer-specific survival. Our study highlights the need to better characterize the immune microenvironment of IDC-P and evaluate its involvement in the poor prognoses of men with IDC-P.

## Figures and Tables

**Figure 1 cancers-15-02217-f001:**
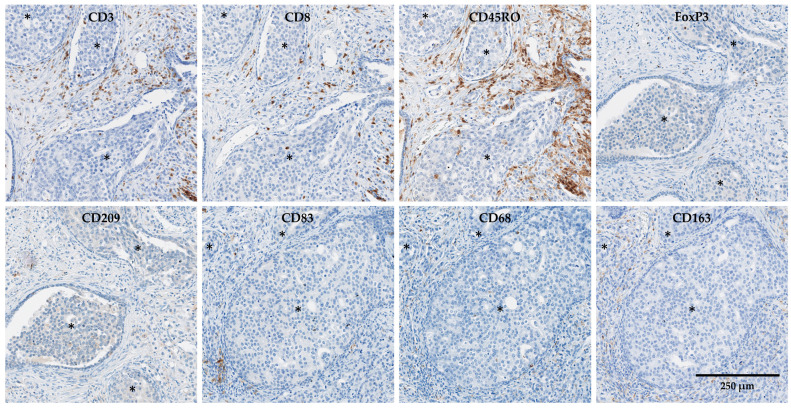
Example of immunohistochemical staining for CD3, CD8, CD45RO, FoxP3, CD209, CD83, CD68 and CD163 in cancer, including intraductal carcinoma of the prostate (IDC-P, asterisks), of the same tissue block. Scale bar: 250 μm.

**Figure 2 cancers-15-02217-f002:**
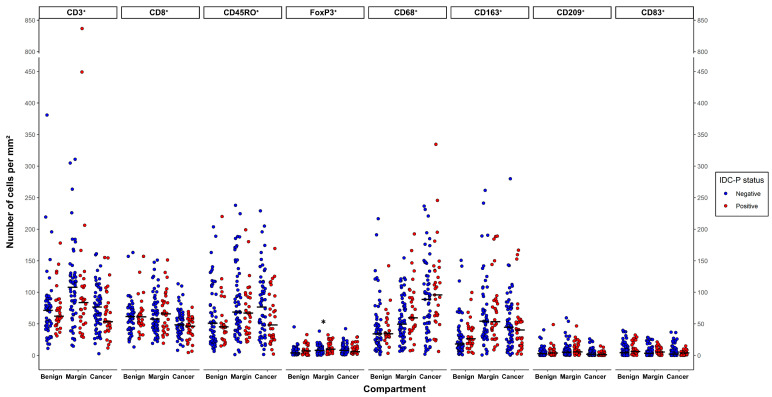
Leucocyte densities depending on IDC-P status. Dots represent individual patients. Horizontal bars represent medians. For each compartment, mean ranks according to IDC-P status were compared using the Mann–Whitney *U* test. * *p* = 0.016.

**Figure 3 cancers-15-02217-f003:**
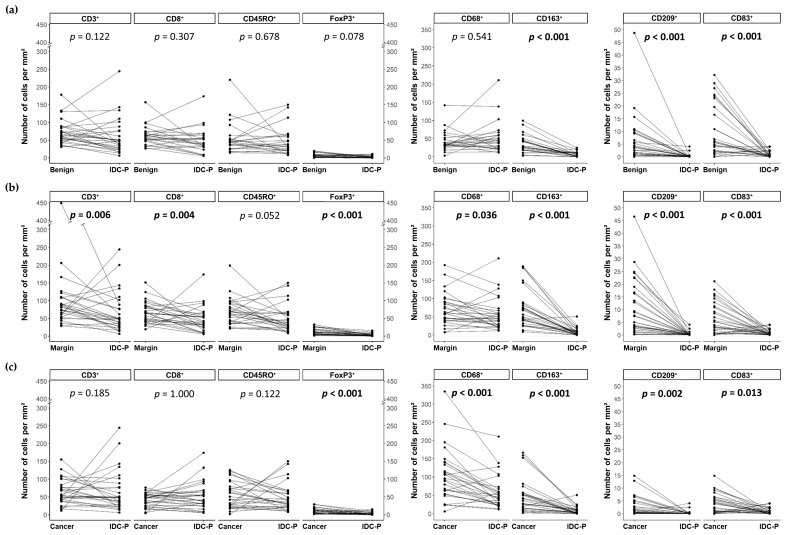
Parallel coordinate plots showing the changes in cell density for each IDC-P-positive patient between the IDC-P compartment and benign tissues (**a**), margins (**b**) and cancer (**c**). Paired-samples sign tests were performed. Bold entities indicate statistically significant *p*-values. IDC-P: intraductal carcinoma of the prostate.

**Figure 4 cancers-15-02217-f004:**
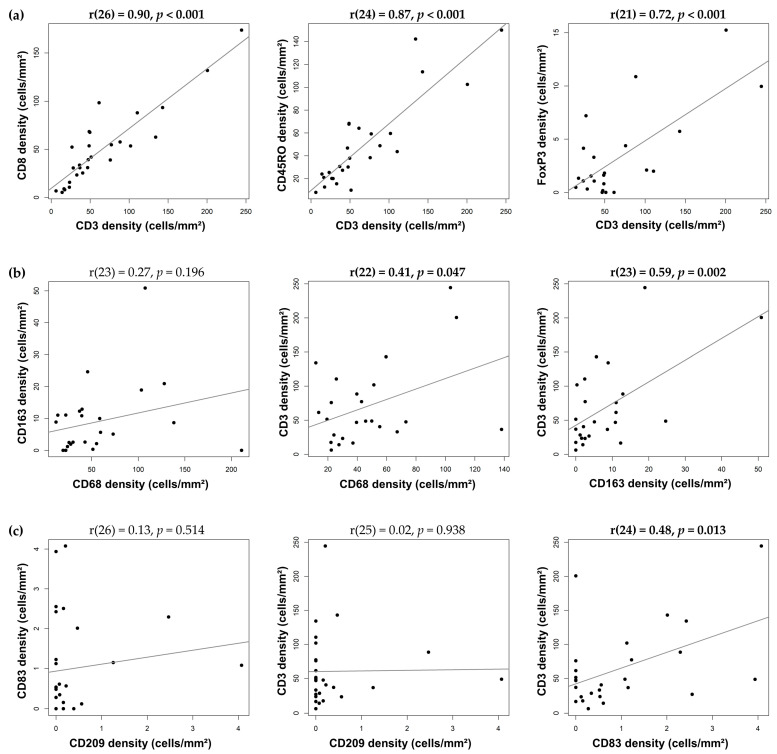
Linear regression and Pearson correlation for cell densities between the different cell types in IDC-P. Correlation between T cells and T cell subtypes (**a**), macrophages and T cells (**b**) and dendritic cells and T cells (**c**). Dots represent individual patients. Bold entities indicate statistically significant correlation.

**Figure 5 cancers-15-02217-f005:**
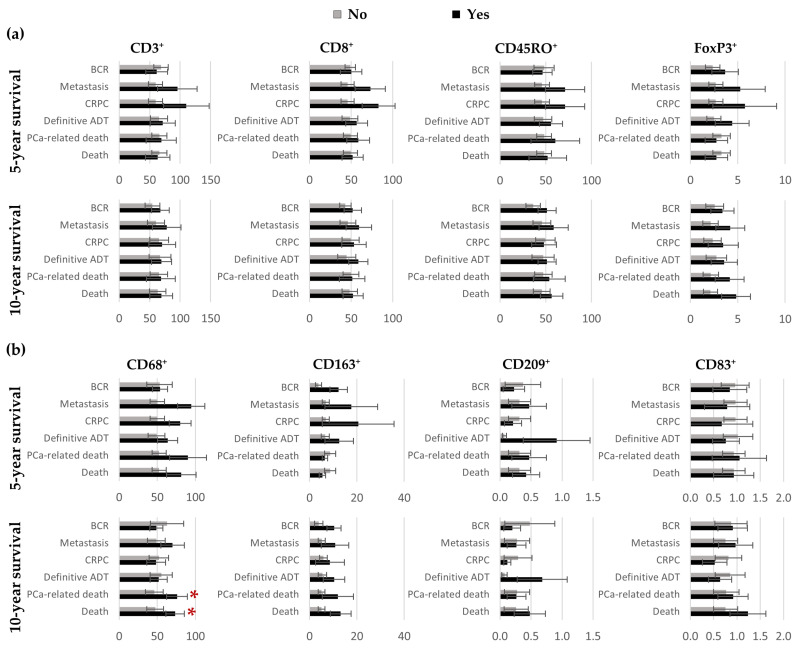
Bar charts representing the mean number of positive T cells (**a**) and antigen-presenting cells (**b**) in the total IDC-P according to patients’ outcomes five and ten years post-surgery. Error bars represent standard errors. For each cell marker, mean ranks according to survival status were compared using the Mann–Whitney *U* test and Benjamini–Hochberg corrections were applied to control for multiple comparisons. * Statistically significant *p*-values. ADT: androgen-deprivation therapy; BCR: biochemical recurrence; CRPC: castration-resistant prostate cancer; IDC-P: intraductal carcinoma of the prostate; PCa: prostate cancer.

**Figure 6 cancers-15-02217-f006:**
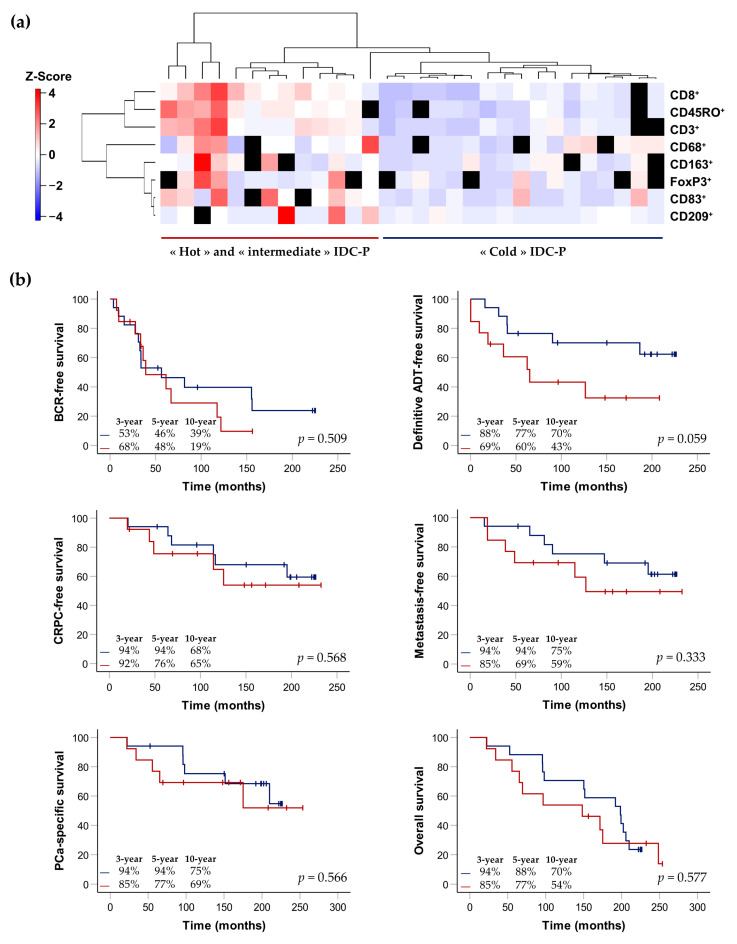
Hierarchical clustering of the eight immune-cell markers according to average cell densities in the total IDC-P (**a**) and Kaplan–Meier curves exploring survival rates of patients with immunologically “cold” IDC-P (blue) and patients with immunologically “hot/intermediate” IDC-P (red) (**b**). (**a**) On the left side: cell-density scale from dark blue (low) to dark red (high). Each row corresponds to a patient and missing slides/quantification data are represented by black squares. The IDC-P-positive patients with more than three missing quantification data were excluded, bringing the total number of patients to 30. (**b**) The *p*-values were calculated using the log-rank test. Three-, five- and ten-year survival rates are indicated in the bottom left of each graph. ADT: androgen-deprivation therapy; BCR: biochemical recurrence; CRPC: castration-resistant prostate cancer; IDC-P: intraductal carcinoma of the prostate; PCa: prostate cancer.

**Figure 7 cancers-15-02217-f007:**
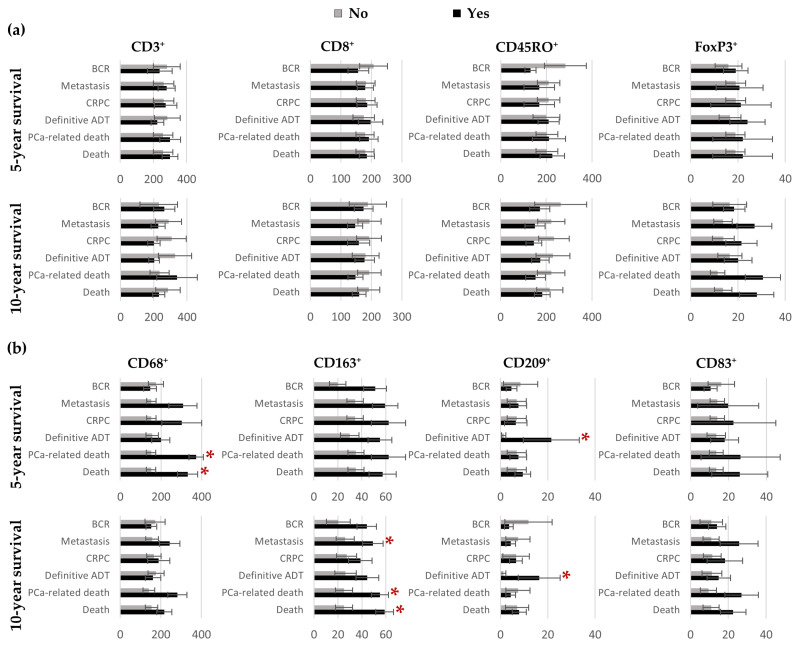
Bar charts representing the mean number of positive T cells (**a**) and antigen-presenting cells (**b**) in IDC-P-immune hotspots according to patients’ outcomes five- and ten-years post-surgery. Error bars represent standard errors. For each cell marker, mean ranks according to survival status were compared using the Mann–Whitney *U* test and Benjamini–Hochberg corrections were applied to control for multiple comparisons. * Statistically significant *p*-values. ADT: androgen-deprivation therapy; BCR: biochemical recurrence; CRPC: castration-resistant prostate cancer; IDC-P: intraductal carcinoma of the prostate; PCa: prostate cancer.

**Figure 8 cancers-15-02217-f008:**
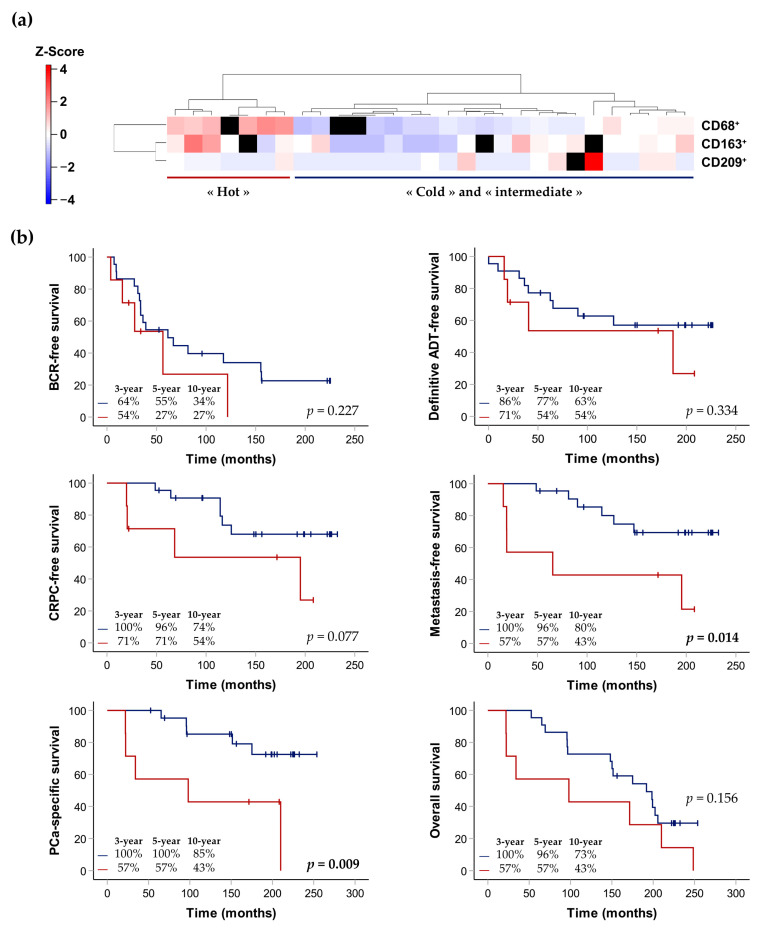
Hierarchical clustering of the three selected immune-cell markers according to cell densities in hotspots (**a**) and Kaplan–Meier curves exploring survival rates of patients with immunologically “cold/intermediate” IDC-P (blue) and patients with immunologically “hot” IDC-P (red), depending on the density in hotspots (**b**). (**a**) On the left side: cell-density scale from dark blue (low) to dark red (high). Each row corresponds to a patient and missing slides/quantification data are represented by black squares. One IDC-P-positive patient with two missing quantification data was excluded, bringing the total number of patients to 29. (**b**) The *p*-values were calculated using the log-rank test. Three-, five- and ten-year survival rates are indicated in the bottom left of each graph. ADT: androgen-deprivation therapy; BCR: biochemical recurrence; CRPC: castration-resistant prostate cancer; IDC-P: intraductal carcinoma of the prostate; PCa: prostate cancer.

**Table 1 cancers-15-02217-t001:** Primary antibodies used in this study.

Antibody	Clone	Company	Dilution	Incubation Time
Anti-CD3	SP7	Abcam, Toronto, ON, Canada	1/500	Overnight
Anti-CD8	4B11	Novocastra, Newcastle upon Tyne, England	1/600	Overnight
Anti-CD45RO	UCHL-1	Abcam, Toronto, ON, Canada	1/6000	Overnight
Anti-FoxP3	236A/E7	Abcam, Toronto, ON, Canada	1/600	Overnight
Anti-CD68	KP1	Abcam, Toronto, ON, Canada	1/800	2 h
Anti-CD163	2G12	Abcam, Toronto, ON, Canada	1/2000	1 h
Anti-CD209	120612	R&D Systems, Minneapolis, MN, USA	1/80	Overnight
Anti-CD83	1H4B	Novocastra, Newcastle-upon-Tyne, UK	1/200	Overnight

**Table 2 cancers-15-02217-t002:** Clinicopathological characteristics of patients according to the presence or absence of intraductal carcinoma of the prostate (IDC-P) on representative slides.

Characteristics	Total Cohort	IDC-P Status		
		Negative	Positive	*p*-Value
	*n* = 96 (%)	*n* = 63 (%)	*n* = 33 (%)	
Mean age at diagnosis (SD)	63.5 (5)	62.8 (5)	64.9 (5)	0.064 ^a^
Mean pre-operative PSA (SD)	13.4 (14)	13.6 (16)	12.9 (10)	0.889 ^b^
Stage pT, *n* (%)				**<0.001 ^b^**
pT2	32 (33)	26 (41)	6 (18)	
pT3a	32 (33)	24 (38)	8 (24)	
pT3b	30 (31)	13 (21)	17 (52)	
pT4	2 (2)	0	2 (6)	
Grade group, *n* (%)				**0.001 ^c^**
1	45 (47)	37 (59)	8 (24)	
2	24 (25)	14 (22)	10 (30)	
3	9 (9)	6 (10)	3 (9)	
4	10 (10)	3 (5)	7 (21)	
5	8 (8)	3 (5)	5 (15)	
Lymph-node involvement, *n* (%)	27 (28)	13 (21)	14 (42)	**0.032 ^d^**
Lymphovascular invasion, *n* (%)	25 (26)	14 (22)	11 (33)	0.332 ^d^
Positive margins, *n* (%)	75 (78)	49 (78)	26 (79)	1.000 ^d^
Biochemical recurrence, *n* (%)	46 (48)	22 (35)	24 (73)	**<0.001 ^d^**
Castration-resistant PCa, *n* (%)	17 (18)	5 (8)	12 (36)	**<0.001 ^d^**
Metastasis, *n* (%)	17 (18)	4 (6)	13 (39)	**<0.001 ^d^**
PCa-related death, *n* (%)	14 (15)	3 (5)	11 (33)	**<0.001 ^e^**
Overall death, *n* (%)	50 (52)	26 (41)	24 (73)	**0.005 ^d^**
Median follow-up in years (IQR)	15.5 (10–19)	15.8 (10–19)	14.3 (8–18)	0.105 ^b^

IDC-P: intraductal carcinoma of the prostate; SD: standard deviation; PSA: prostate-specific antigen; PCa: prostate cancer; IQR: inter-quartile range. Bold entities indicate statistically significant *p*-values. ^a^ Independent-samples *t*-test, ^b^ Mann–Whitney *U* test; ^c^ Welch’s *t*-test; ^d^ Pearson’s chi-square; ^e^ Fisher’s exact test.

**Table 3 cancers-15-02217-t003:** Association of clinical and pathological variables with immunologically “cold”, “intermediate” or “hot” IDC-P.

Characteristics	Total IDC-P			CD68/CD163/CD209 Hotspots		
	Cold *n* = 17	Hot and Intermediate *n* = 13	*p*-Value	Cold and Intermediate *n* = 22	Hot *n* = 7	*p*-Value
Mean age at diagnosis (SD)	64.6 (5)	65.5 (6)	0.650 ^a^	65.9 (5)	61.7 (7)	0.149 ^a^
Mean pre-operative PSA (SD)	11.4 (9)	13.5 (10)	0.476 ^a^	13.4 (10)	8.7 (4)	0.524 ^a^
Stage pT *, *n* (%)			1.000 ^a^			0.162 ^a^
pT2	2 (12)	3 (23)		4 (8)	1 (14)	
pT3a	6 (35)	2 (15)		8 (36)	0 (0)	
pT3b-pT4	9 (53)	8 (62)		10 (46)	6 (86)	
Grade group *, *n* (%)			0.575 ^a^			0.182 ^a^
1–2	10 (59)	6 (46)		14 (64)	2 (29)	
3	1 (6)	1 (8)		1 (5)	1 (14)	
4–5	6 (35)	6 (46)		7 (32)	4 (57)	
Lymph-node involvement, *n* (%)	6 (35)	7 (54)	0.460 ^b^	8 (36)	4 (57)	0.403 ^c^
Lymphovascular invasion, *n* (%)	5 (29)	5 (39)	0.694 ^c^	6 (27)	3 (50)	0.352 ^c^
Positive margins, *n* (%)	13 (77)	12 (92)	0.355 ^c^	17 (77)	7 (100)	0.296 ^c^
Extraprostatic extension, *n* (%)	15 (88)	7 (54)	**0.049 ^c^**	17 (77)	4 (57)	0.357 ^c^
Seminal vesicle invasion, *n* (%)	10 (59)	8 (62)	1.000 ^b^	11 (50)	6 (86)	0.187 ^c^
Median follow-up in years (IQR)	16.5 (8–18)	12.4 (5–16)	0.263 ^a^	15.3 (8–19)	8.2 (2–17)	0.149 ^a^

IDC-P: intraductal carcinoma of the prostate; SD: standard deviation; PSA: prostate-specific antigen; PCa: prostate cancer; IQR: inter-quartile range. * pT3b–pT4 stages and grade groups 1–2 and 4–5 were combined because of the small number of patients. Bold entities indicate statistically significant *p*-values. ^a^ Mann–Whitney *U* test; ^b^ Pearson’s chi-square test; ^c^ Fisher’s exact test.

## Data Availability

The data presented in this study are available on request from the corresponding author. The data are not publicly available due to ethical restrictions.

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
