# Peer review of "Leukocytic Infiltration of Intraductal Carcinoma of the Prostate: An Exploratory Study"

_cancers, 2023, doi:10.3390/cancers15082217_

Round 1

Reviewer 1 Report

This is a well written somewhat lengthy report of an interesting pathological finding of potential immunologic relevance in in prostate cancer by a group with a remarkable track record. In an era where immunotherapy faces considerable changes in this particular tumor entity data like this even if they are exploratory are of considerable interest for the clinical community.   

Author Response

We greatly appreciate your review of our manuscript, thank you for your kind words.

Reviewer 2 Report

In this work, Diop and colleagues aim to identify a clinical significance of tumor infiltrating leukocytes in intraductal prostate cancer (IDC-P), a particularly aggressive histological subtype of prostate cancer (PCa), associated with PCa-related death and low response to standard treatment.

The authors analyzed by immunohistochemical staining the presence of different subpopulations of immune cells and no differences were observed between tumors with negative and positive IDC areas.

Subsequently, the authors focused on IDC-P positive patients (n=33), analyzing in depth the amount and localization of the different subtypes of immune infiltrate.

Overall, they observed that FoxP3+ regulatory T cells, CD68+CD163+ macrophages, and CD209+CD83+ mature dendritic cells were less abundant in IDC-P than in adjacent normal and cancerous tissue.

Then the authors divided the IDC-P positive patients according to the expression of immune markers into hot/intermediate (medium/high expression of several markers) and cold (generally low expression of markers) IDC-P.

They compared the clinical outcomes (no BCR, no metastasis, overall survival…) between warm/intermediate IDC-P and cold IDC-P. No statistical differences were observed for all parameters considered.

The authors concluded that further studies are needed to better characterize the immune microenvironment of IDC-P and evaluate its involvement in the poor prognosis of men with IDC-P. The conclusion of the work is therefore that the authors themselves realize that the results shown in this study are not able to determine whether or not there is a correlation between immune infiltrate and the course of the disease.

The authors highlight that the main limitation of the study is the small number of cases studied.

This is absolutely true. From the results shown it is not even possible to understand whether there is a trend in the correlation between the immune system, cell subtypes, and disease.

The idea of ​​finding a correlation between this is interesting, but it is not developed in the least in this work.

Probably the small number of patients does not allow it, but it could be interesting and useful to diversify patients according to the type of cells that increase in the tissue.

Commonly, Foxp3 regulatory T cells and CD8+ cytotoxic T cells usually have opposite significance in the progression of tumor diseases. Putting effector cells and suppressor cells together may be why we don't see differences between the groups considered.

Could the authors try to classify patients by other parameters to look for a potential correlation with disease progression?

Author Response

We thank the reviewer for his suggestions. We added the results of the association of the mean number of positive T cells and antigen-presenting cells in the total IDC-P with patients’ outcome five- and ten-years post-surgery (Figure 5 in the revised manuscript) and combined former figures 5 and 6 (Figure 6 in the revised manuscript).

Furthermore, we added the results of the the association of the densities of positive T cells and antigen-presenting cells in IDC-P immune hotspots with patients’ outcome five- and ten-years post-surgery (Figure 7 in the revised manuscript). We found that higher expression of CD68, CD163 and CD209 in hotspots were associated with poor outcome and generated a heatmap with the three cell types to separate patients in two groups according to cell densities: a “cold/intermediate” group and a “hot” group (Figure 8a in the revised manuscript). Survival analysis showed that higher expression of CD68, CD163 and CD209 are associated with the development of metastasis and prostate cancer-specific death (Figure 8b in the revised manuscript). Results of the association of clinical and pathological variables depending on the two groups were also added to Table 3.

The abstract (lines 41-44 in the revised manuscript), the introduction (lines 85-89 in the revised manuscript), the methods section (lines 143-145, 167-168 and 173 in the revised manuscript), the results (lines 281-400 in the revised manuscript), the discussion (lines 408-415, 492-499 in the revised manuscript) and the conclusion (lines 541-544 in the revised manuscript) were modified accordingly to include the results of the immune infiltrate in immune hotspots.

Moreover, we added sentences in the discussion stating that IDC-P is a focal lesion that represent less than 5% of the tumor volume in most cases, explaining the difficulties in accessing samples and the absence of data on the immune infiltrate of IDC-P (lines 516-523 in the revised manuscript).

The revised text reads as follows:

“The main limitation is the small number of patients with IDC-P in our cohort (n = 33). However, IDC-P is often focal, with more than 75% of the specimens with IDC-P harboring IDC-P in less than 5% of the full tumor volume [56]. Furthermore, the evaluation of the inflammatory infiltrates of adjacent invasive carcinoma, tumor margins and benign glands requires spatial localization which cannot be provided by biopsies and TMAs. These limitations in sample availability could explain why the immune infiltrate of IDC-P had not yet been described. Our findings must still be replicated in larger independent cohorts, but our work was essential to begin to unravel the signification of immune infiltrates in IDC-P.”

56 Trudel, D.; Downes, M.R.; Sykes, J.; Kron, K.J.; Trachtenberg, J.; van der Kwast, T.H. Prognostic impact of intraductal carcinoma and large cribriform carcinoma architecture after prostatectomy in a contemporary cohort. Eur J Cancer 2014, 50, 1610-1616, doi:https://doi.org/10.1016/j.ejca.2014.03.009.